# Urgent Transcatheter Edge-to-Edge Repair for Severe Mitral Regurgitation in Patients with Refractory Cardiogenic Shock

**DOI:** 10.3390/jcm11195617

**Published:** 2022-09-23

**Authors:** Nimrod Perel, Elad Asher, Luoay Taha, Nir Levy, Yoed Steinmetz, Hani Karameh, Mohammad Karmi, Tomer Maller, Emanuel Harari, Danny Dvir, Michael Glikson, Shemy Carasso, Mony Shuvy

**Affiliations:** 1Jesselson Integrated Heart Centre, Shaare Zedek Medical Center and Faculty of Medicine, Hebrew University, 12, Shmu’el Bait, P.O. 3235, Jerusalem 9103102, Israel; 2The Azrieli Faculty of Medicine, Bar Ilan University, Zefat 1311502, Israel

**Keywords:** TEER, MR, CS

## Abstract

Introduction Patients suffering from cardiogenic shock (CS) and mitral regurgitation (MR) demonstrate worse prognosis, with higher mortality rates. We sought to evaluate the effectiveness of urgent valve intervention of the mitral valve, using transcatheter edge-to-edge repair (TEER) procedures in patients presenting with CS in a tertiary Intensive Coronary Care Unit (ICCU). Methods and Results Patients with unremitting CS and severe MR were selected for urgent TEER. Baseline clinical and echocardiographic characteristics were recorded, as well as procedural success (MR severity and hemodynamics), and 30-days and 6-month mortality. Urgent TEER was done in 13 patients, whose average age was 70 years; 12 (92%) of the patients were male. All 13 patients had suffered previous ischemic heart disease—12 (92%) with either acute severe MR or worsening of previously known MR by an acute ischemic event. Using the SCAI criteria, 8 patients (61%) were classified as ‘E’ (Extreme) category; 4 (31%) were classified as ‘C’. At 30 days, 12 out of the 13 patients survived (corresponding to an 8% mortality rate); all of those 12 patients remained alive at 6 months post-admission/procedure. Conclusions The use of TEER was associated with greater 30-day and 6-month survival rates, compared to the worldwide mortality rates of patients admitted with CS. This finding may change the previous paradigm that CS and MR are associated with the worst outcome, and we might be able to offer these patients a safe and effective therapeutic option.

## 1. Introduction

Acute and subacute severe mitral regurgitation (MR) is a known etiology of cardiogenic shock (CS). The prognosis of patients with CS and MR is usually grim, with no therapy directed at the underlying pathology [1].

The main etiologies of acute MR include ruptured chordae tendineae of a degenerated valve, infective endocarditis damaging leaflets, papillary muscle rupture due to ischemia or acute myocardial infarction with left ventricular dysfunction and adverse remodeling causing systolic leaflet tethering. In contrast to chronic MR, the pathophysiologic process of acute MR is a sudden volume and pressure overload overwhelming the left ventricle. The ventricle does not have the time to adapt and compensate for the large regurgitant volume. This significant backflow of blood impairs the forward flow, leading eventually to shock. Medical treatment, including pharmacological or mechanical afterload reduction with intra-aortic balloon pump, can provide temporary stabilization. To break this vicious cycle, however, the structural damage of the valve usually needs to be addressed. The current main therapeutic option is surgical correction of the valve, but many of these patients are at very high surgical risk, related to both the CS status and previous illnesses, and are deemed inoperable [2,3].

Transcatheter edge-to-edge repair (TEER) has emerged in the last decade, as a potential therapeutic intervention for MR. Current guidelines recommend the use of TEER of the mitral valve in the setting of chronic regurgitation in stable patients, yet there is no clear recommendation for TEER intervention in patients with CS [4,5,6].

We aimed to evaluate the outcome of patients in CS who underwent urgent TEER procedures.

## 2. Methods

We collected the data of patients admitted to Shaare Zedek Medical Center between January 2020 and July 2021. The analysis included those who presented with CS and concomitant severe MR, who underwent urgent TEER. Patients were selected to undergo the procedure, due to refractory shock that was not responsive to standard therapy with concomitant severe MR. Cardiogenic shock was diagnosed based on systolic blood pressure < 90 mmHg, with clinical and laboratory signs of hypoperfusion, such as oliguria, cold extremities and elevated lactate levels. The severity of the CS was defined according to the criteria published by the Society of Cardiovascular Angiography and Intervention (SCAI) [7]. MR severity was diagnosed and evaluated with transthoracic echocardiography (TTE) and transesophageal (TEE). These were used for further evaluation of valve anatomy and feasibility assessment for TEER. Evaluation of severe MR was based on the current American Society of Echocardiography guidelines, and was graded from 1 to 4, as mild, moderate, moderate-severe or severe [8,9]. All patients found suitable for TEER were at high risk for cardiovascular surgery. They were all assessed by the departmental Heart team, and the procedure was approved for them.

Transcatheter edge-to-edge repair was performed by implantation of a ‘MitraClip^®^’ device (Abbott, Abbott Park, IL, USA). The device system was delivered to the left atrium via transeptal puncture. The procedure was done with fluoroscopic and TEE guidance. TEE was used for grading the severity of the MR, and for decision-making during procedure (implantation of additional clip or ending the procedure). At least one clip was implanted in all patients. Additional clips were implanted to achieve a more significant MR reduction, when deemed necessary by the operators. The left atrial V-wave was measured and recorded, before and after TEER implantation. Measurement after the implantation was done while the TEE probe was in still in place, and before the iatrogenic shunt was unblocked (removal of the clip delivery system), to provide more reliable measurements of LA pressures. We calculated the absolute reduction in the V-wave before and after the procedure.

The patients underwent transthoracic echocardiographic examination on the day following the procedure, and the severity of the MR was recorded using the same parameters as before the procedure.

Thirty-day and 6-month mortality rates were recorded. The New York Heart Association (NYHA) functional class was obtained at follow-up visit, post-discharge.

The results are expressed as the mean standard deviation, or as a percentage. The Wilcoxon signed-rank test was used to compare the same group at two different time-points before and after the procedure; it is a non-parametric test of paired observation, that does not assume normal distribution of the observed measurements.

## 3. Results

Thirteen patients who presented with CS and concomitant severe MR underwent urgent TEER. The mean age was 70.3 ± 10.3 years, and 12 (92%) of the patients were men (Table 1).

All patients presented with an acute ischemic event (STEMI or NSTEMI), and 11 underwent PCI according to current recommendations, while the remaining 2 patients with an NSTEMI diagnosis had either undergone later revascularization or no revascularization [10,11]. Of the 13 patients included in the study, 8 were diagnosed with STEMI, and 5 were diagnosed with NSTEMI. From the STEMI group, 3 of the 8 patients suffered from acute stent thrombosis, 2 had late arrival presentation and 1 suffered from an acute occlusion of the LIMA graft. Overall, in most patients, the ischemic MR developed after a large myocardial infarction, despite successful revascularization, or in patients with incomplete or partially successful revascularization. All but 1 patient had MR due to leaflet tethering related to their infarct territory designated as secondary MR. One patient had a flail leaflet, and was designated as primary MR (Table 1).

Eight patients (61%) were classified as E (Extreme) category, based on the Society for Cardiovascular Angiography and Interventions (SCAI) criteria (Figure 1).

All the patients experienced improvement in MR severity grading by at least two grades the day after the TEER, and 7 patients (54%) experienced improvement from severe or moderate-severe to mild (*p* = 0.001) (Figure 2A). Pre-procedural and post-procedural echocardiographic parameters are disclosed in Table 2, showing improvement in the forward flow (continuity calculate stroke volume), decrease in the calculated mitral regurgitant volume and MR jet vena contracta. Notably, the LV 2D biplane dimensions and stroke volume did not change. The average V-wave before the procedure was 36.8 ± 9 mmHg, and was reduced to an average of 18.7 ± 7 mmHg (*p* = 0.002) (Figure 2B). Most of the patients (84%) showed early clinical improvement, with 11 patients (84%) weaned off either mechanical ventilation or vasopressors support within 48 h. The median of pre-TEER length of stay in the CCU was 14 days (IQR 8–26), compared to the post-TEER stay in the CCU, which was 7 days (IQR 4–20).

Twelve out of the 13 patients survived at 30 days, a mortality rate of 8%; all 12 of those patients survived at 6 months. The only patient that did not survive at 30 days died due to septic shock, 2 weeks post-intervention (Figure 2).

At follow-up visit, 9 out of the 12 patients (75%) were NYHA II, while the other 3 patients (25%) were NYHA III.

## 4. Discussion

We have shown here that Mitral TEER was feasible and effective in the treatment of patients with cardiogenic shock with severe MR admitted to the CCU in the course of an acute coronary event. In the short-stay, the 1-month mortality rate was 8%, with survival extending to 6 months after procedure. The long pre-CCU length of stay emphasized the gravity of pre-TEER status, and the time it took the staff to be convinced that MR severity hampered hemodynamic improvement, and to then suggest TEER as the preferred mode of action. The shorter and post-TEER CCU stays suggest that TEER did change the clinical course of our patients, enabling weaning from medical and mechanical support, and discharge from the CCU.

The current recommended treatment for CS and severe MR includes medical treatment, mechanical support and, in some cases, mitral valve surgery. However, patients are often deemed inoperable, due to their severe clinical status. This inherent limitation in the current treatment of CS and MR emphasizes the important role of less invasive options, such as percutaneous repair. Recent data suggest that urgent intervention in the acute setting is safe and effective, and is associated with improved outcome [12,13].

The MITRA-SHOCK study enrolled 31 patients, with an average age of 73, and had 21.6% and 55.8% mortality rates at 30 days and 6 months, respectively [14]. Based on current literature, the average 30-day mortality rate in CS patients is around 50%; these rates have not significantly improved, despite improvement in care and technology [15,16]. The mortality rate of patients in all forms of CS at our institute during that time was 44%.

Although 60% of patients enrolled in our analysis were in the most severe form of CS, according to SCAI criteria, the 30-day mortality rate was only 8%. Importantly, clinical improvement persisted during follow-up, and all 12 patients that survived at 1 month also survived at 6 months (100%). We also observed a positive effect on quality of life, with a decrease in repeat hospitalizations, suggesting that successful TEER (reflected by MR reduction and decrease in the V-wave) can provide a bridge to recovery, reversing the vicious cycle maintaining shock and pulmonary edema; depending on the sustainability of procedural results, it may well serve as the definitive procedure for these patients. In certain cases, TEER may also serve as a bridge to surgery, reducing its risk in clinically recovering patients.

As seen in previous studies, we also faced a difficulty in post-TEER echocardiographic assessment of MR severity. Proximal isovelocity surface area (PISA) assessment was mostly unavailable post-procedure, which called for a multi-method assessment of MR (vena contracta, mixed mode volumetric assessments and pulmonary venous flow patterns) (Table 2), as suggested by recent recommendations [9].

## 5. Limitations

This analysis was based on retrospective observational data without a control group, and was limited in size. Patients that were not selected for this procedure could not serve as controls, as they either did not have severe enough MR (and were therefore not comparable) or were in a worse clinical state, with prohibitive risk even for this procedure (again, not comparable). The small sample-size limited us from making a global conclusion regarding this group of patients. A selection bias may have prevented us from including all patients with CS and severe MR, yet we did do our best to include all consecutive patients fulfilling prior publications. Finally, the comparison of mortality rates with the current reported rates in the literature was limited, due to unmeasured confounders.

## 6. Conclusions

Thirty-day and 6-month mortality rates were relatively low among patients with CS and severe MR who underwent acute TEER.

This finding may change the previous paradigm that CS and MR are associated with the worst outcome, and may offer a safe and effective therapeutic option.

## Figures and Tables

**Figure 1 jcm-11-05617-f001:**
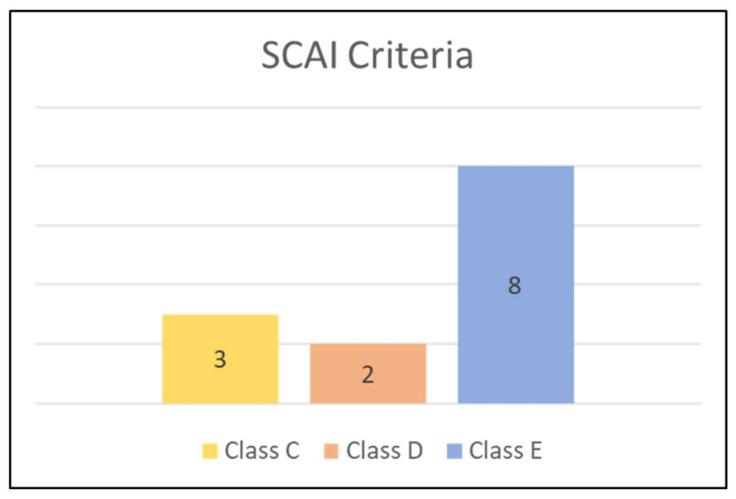
Patients’ severity, based on the Society for Cardiovascular Angiography and Interventions (SCAI) criteria, according to the SCAI classification of cardiogenic shock.

**Figure 2 jcm-11-05617-f002:**
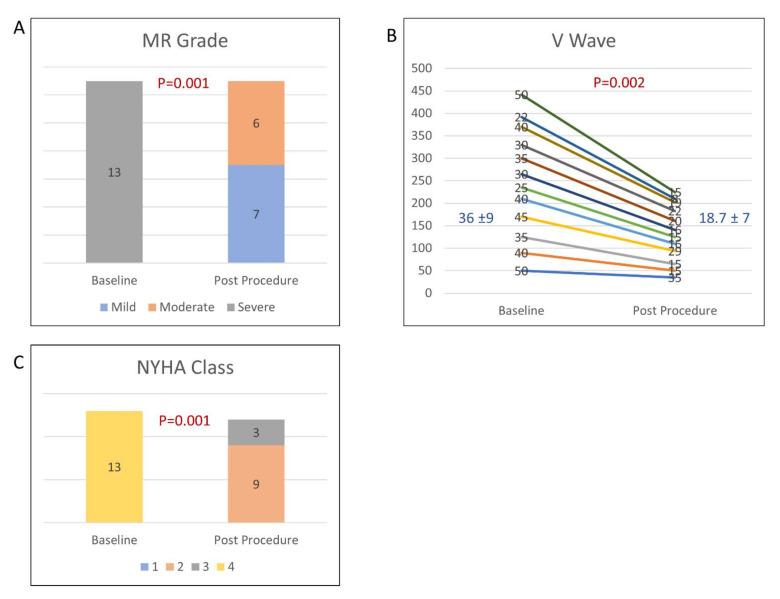
The impact of TEER procedure on: (**A**) severity of mitral regurgitation (MR); (**B**) left atrial V-wave; (**C**) NYHA class.

**Table 1 jcm-11-05617-t001:** Baseline patient characteristics, procedure outcome and follow-up data.

	Baseline																Post-Procedure				
Patient #	Age	Sex	MR Grade	V-Wave pre	MR TYPE	HTN	HPL	DM	AF	CKD	IHD	SCAI	EuroScore II	Mechanical Ventilation	Vasopressors	MCS	30D Mortality	6M Mortality	V-Wave after Clip	MR Grade 1 D Post-Procedure	NYHA
1	80	M	Severe	50	Secondary	YES	YES	YES	YES	YES	YES	C	37	NO	Nor	IABP	Alive	Alive	35	Moderate	2
2	90	M	Severe	40	Secondary	YES	NO	NO	NO	YES	YES	C	35	NO	No	No	Alive	Alive	15	Moderate	2
3	74	M	Severe	35	Secondary	YES	YES	YES	NO	NO	YES	D	27	YES	Nor	IABP	Alive	Alive	15	Moderate	2
4	82	M	Severe	45	Secondary	YES	NO	NO	YES	NO	YES	E	27	YES	Nor+Phen	No	Alive	Alive	29	Moderate	2
5	62	M	Severe	40	Secondary	NO	YES	NO	NO	NO	YES	E	16	YES	Nor+Dobu+Phen	Impella+IABP	Alive	Alive	16	Mild	3
6	71	F	Severe	25	Secondary	NO	NO	NO	NO	NO	YES	C	23	NO	Nor	No	Alive	Alive	15	Mild	2
7	63	M	Severe	30	Secondary	YES	NO	NO	YES	NO	YES	E	16	YES	Noe+Phen	IABP	Dead	Dead	16	Moderate	-
8	64	M	Severe	35	Secondary	YES	YES	NO	YES	NO	YES	E	16	YES	Nor+Dob	No	Alive	Alive	20	Mild	2
9	78	M	Severe	N/A	Secondary	YES	YES	YES	NO	NO	YES	D	29	NO	Nor+Phen	No	Alive	Alive	N/A	Mild	2
10	51	M	Severe	30	Secondary	NO	NO	NO	NO	NO	YES	E	15	YES	Nor	IABP	Alive	Alive	22	Mild	3
11	66	M	Severe	40	Primary	YES	NO	YES	NO	NO	YES	E	20	YES	Nor	IABP+ECMO	Alive	Alive	19	Mild	3
12	66	M	Severe	22	Secondary	NO	YES	NO	NO	NO	YES	E	17	YES	Nor+Dob	IABP+ECMO	Alive	Alive	8	Mild	2
13	67	M	Severe	50	Secondary	YES	YES	NO	NO	NO	YES	E	18	YES	Nor	IABP	Alive	Alive	15	Moderate	2
**Mean/Rate**	**70**	**92%**	**100%**	**36.8**	**92%**	**69%**	**54%**	**31%**	**31%**	**15%**	**100%**	**62%**	**23**	**77%**	**85%**	**61%**	**8%**	**8%**	**18.8**	**54%**	**2.3**
		Male	severe		Functional							E Category						Mortality		mild	

MR, mitral regurgitation; HTN, hypertension; HPL, hyperlipidemia; DM, diabetes mellitus; AF, atrial fibrillation; CKD, chronic kidney disease; NOR, noradrenaline; PHEN, phenylephrine; DOB, dobutamine; MCS, mechanical circulatory support; IABP, intra-aortic balloon pump; ECMO, extracorporeal membrane oxygenation; IHD, ischemic heart disease; NYHA—New York Heart Association heart-failure functional class.

**Table 2 jcm-11-05617-t002:** Echocardiographic parameters before and after TEER: (a) echocardiographic parameter before TEER; (b) echocardiographic parameter after TEER.

**Patient #**	LVEDD (cm)	LVESD (cm)	LVEDV Biplane (mL)	LVESV Biplane (mL)	LVSV Biplane (mL)	EF Biplane (mL)	LVSV Continuity (mL)	EROA (cm^2^)	Regurgitant Volume (mL) *	Vena Contracta (mm)	PV Flow Pattern	LASd (cm)
**1**	5.4	3.1	100	35	65	53%	53	0.41	42	6		5.1
**2**	6.7	5	116	72	44	38%		0.59	65	9	S reverse	5.5
**3**	5.7	4.8	204	154	131	64%	39	0.28	41	9		5.9
**4**	6.1	5	195	118	77	39%	40		37	8	S reverse	4.9
**5**	5.6	4.5	127	72	55	43%	47	0.28	58	8		4.8
**6**	5.0	4.2	160	107	53	33%	37	0.25	40	9	S reverse	4.3
**7**	6.3	5	237	153	84	35%	63	0.53	56	7	S Blunting	5.3
**8**	5.7	3.5	169	112	57	34%	34		23	5	S Blunting	4.9
**9**	6.0	5.3	144	90	54	38%	34		20	7	S Blunting	3.8
**10**	6.0	5.1	162	104	58	36%	34	0.27	29	12	S Blunting	5.4
**11**	5.6	5.2	142	90	52	37%	35		27	10	S Blunting	5.1
**12**	5.5	4.7	106	52	54	51%	28		26	6	S reverse	4.1
**13**	4.7	3.4	140	55	85	61%	44	0.64	83	8	S Blunting	5.3
**Average**	5.7 ± 0.5	4.5 ± 0.7	154 ± 40	93 ± 37	67 ± 24	43 ± 11	41 ± 10	0.4 ± 0.2	42 ± 19	8 ± 2		5.0 ± 0.6
**Patient #**	**LVEDD** **(cm)**	**LVESD** **(cm)**	**LVEDV** **Biplane** **(mL)**	**LVESV** **Biplane** **(mL)**	**LVSV** **Biplane** **(mL)**	**EF Biplane** **(mL)**	**LVSV Continuity** **(mL)**	**EROA** **(cm^2^)**	**Regurgitant** **Volume** **(mL)** ******	**Vena** **Contracta** **(mm)**	**PV Flow Pattern**	**LASd** **(cm)**
**1**	4.8	3.3	77	37	40	55%	51		4	3		4.8
**2**	5.5	4.8	100	57	43	43%		0.24	37	4	SD equal	4.2
**3**	5.5	4	227	175	52	23%	42		10	3	SD equal	6
**4**	6.1	50	197	119	78	40%	47		31	5	S Dominance	4.5
**5**	5.4	4.3	156	86	70	45%	53		17	2		4.5
**6**	5.5	4.6	165	91	74	45%	68		6	3		3.7
**7**	6.3	4.7	217	135	82	38%	72		10	3	S Dominance	5.1
**8**	5.1	3.9	146	88	58	40%	57		1	1	S Dominance	4.7
**9**	6.1	5.7	157	115	42	27%	34		8	3	SD equal	4.9
**10**	5.9	4.9	157	84	73	46%	45		28	2	S Blunting	5.2
**11**	6.3	5.3	142	68	74	52%	57		20	2	SD equal	4.2
**12**	5.2	3.9	118	52	66	56%	53		13	2	S Dominance	4.1
**13**	6.1	4.1	146	63	83	57%	72		11	4	SD equal	5.1
**Average**	5.6 ± 0.5	5.0 ± 0.7	154 ± 43	90 ± 38	64 ± 15	44 ± 12	54 ± 11		15 ± 11	3 ± 1		4.7 ± 0.6

Abbreviation: TEER, transcatheter edge-to-edge repair; LVEDD, left ventricle end diastolic diameter; LVESD, left ventricle end systolic diameter; LVEDV, left ventricle end diastolic volume; LVESV, left ventricle end systolic volume; LVSV, left ventricle stroke volume; EF, ejection fraction; EROA, effective regurgitant orifice area; PV, pulmonic vein; LA Sd, left atrium systolic diameter; S, Systolic. * Regurgitant volume was mostly calculated by the PISA (proximal isovelocity surface area) method (EROA calculation available); secondary calculation—mixed-mode volumetric assessment—RV= biplane LVSV-continuity LVSV. ** Regurgitant volume was mostly calculated by mixed-mode volumetric assessment—RV= biplane LVSV-continuity LVSV; PISA was mostly unavailable post-TEER.

## Data Availability

Data is stored at the SZMC TEER registry, please contact the corresponding author for further information.

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
