# Peer review of "Urgent Transcatheter Edge-to-Edge Repair for Severe Mitral Regurgitation in Patients with Refractory Cardiogenic Shock"

_jcm, 2022, doi:10.3390/jcm11195617_

Round 1

Reviewer 1 Report

I read with interest the article entitled “Urgent Transcutaneous Edge to Edge Repair for Severe Mitral Regurgitation in Patients with Refractory Cardiogenic Shock”.

The role of transcatheter edge-to-edge mitral repair with MitraClip as an adjunct therapy in patients with cardiogenic shock remains poorly described. 

In the present study, TEER using the MitraClip system appears to be a viable therapeutic salvage strategy, demonstrating procedural safety and efficacy in mitral regurgitation (MR) reduction in patients with significant MR and cardiogenic shock. Although heterogeneous in MR aetiology, successful device implantation was associated with decreased risk for mortality over the 6-months of follow-up, supporting the hypothesis that MR reduction in the acute phase of shock may represent a therapeutic target addressable using a percutaneous approach. 

The paper is well written, the study is clearly described and, finally, the results are right displayed.

Few considerations have to be made:

-       The data are observational and are thus susceptible to all biases inherent to this study's design

-       The small sample size and the lack of a comparison group make difficult the generalizations of the results

-       Numerous clinical (eg. inotropes treatment, type of mechanical circulatory support) and echocardiographic data pre and post-procedural were missing from the cohort (EROA, vena contracta, left ventricular volumes and diameters, ejection fraction). These data are needed in order to evaluate better the outcomes of TEER in the setting of cardiogenic shock. 

-       Data regarding mortality is deficient if compared with MITRA-shock and another small meta-analysis regarding this topic. Ho do the authors explain such low mortality data compared to the MITRA-shock study and the other studies on this topic?

Author Response

We thank the reviewer for this comment.

We added a new table showing echocardiographic parameters before and after the procedure (attached as a PPT presentation).  In acute MR, the rapid increase of pressure in the left atrium leads to smaller pressure gradient with an earlier pressure equalization. The MR occurs mainly in the earlier part of systole, and a murmur as well as the color jet seen in TEE can be diminished. Because the color doppler often will not show a large turbulent flow disturbance, MR may be underestimated or not appreciated at all. Moreover, most of the doppler methods for assessing MR severity apply to chronic and not to acute severe MR.(1,2) Thus, the validity of MR severity in acute setting was based on the evaluation of chronic MR and the clinical judgment in evaluating patients with acute MR is highly important.  The evaluation of the regurgitant volume post the procedure was based on hemodynamic calculation due to the difficulty in performing the PISA method after the TEER.                                                                                                                                               In addition, we have added a table of the types of vasopressors and inotropic support as well as the type of mechanical circulatory support. This changes are added in table 1

Patient #

Inotropes

MCS

#1

Noradrenaline

IABP

#2

-

no

#3

Noradrenaline

IABP

#4

Noradrenaline +Phenylephrine

no

#5

Noradrenaline +Dobutamine+Phenylephrine

Impella+IABP

#6

Noradrenaline

no

#7

Noradrenaline +Dobutamine

IABP

#8

Noradrenaline +Phenylephrine

no

#9

Noradrenaline

no

#10

Noradrenaline

IABP

#11

Noradrenaline

IABP +ECMO

#12

Noradrenaline +Dobutamine

IABP+ECMO

#13

Noradrenaline

IABP

  1. Data regarding mortality is deficient if compared with MITRA-shock and another small meta-analysis regarding this topic. Ho do the authors explain such low mortality data compared to the MITRA-shock study and the other studies on this topic

Response: We appreciate this important comment.

In the MITRA-SHOCK study they reported survival rate and not mortality rate. This means that the mortality rate was 21.6% at 30 days in comparison to our study mortality rate of 8%. In addition, we believe that the small number of patients in this study can further explain the differences in mortality rates.

We have corrected this segment in the discussion segment:

The MITRA-SHOCK study enrolled 31 patients with an average age of 73 and had 21.6% and 55.8% mortality rates at 30 days and 6 months, respectively.

Reviewer 2 Report

Severe acute mitral regurgitation (MR) is rare but severe and challenging disease that requires emergent care and proper treatment. Although surgical correction is usually indicated for acute MR,    many patients are considered to be at prohibitive surgical risk, because of ongoing cardio-respiratory instability, developing multiorgan-failure, frailty, advanced age or the presence of co-morbidities. Additionally outcomes of conventional surgical interventions have remained poor over time, witha mortality up to 40%. Transcatheter mitral valve edge-to-edge repair (TEER) has now been  widely accepted and used for management of selected patients with chronic MR, who are declined for surgical intervention. Nowadays, percutaneous mitral valve repair with the MitraClip device is receiving increased attention for the treatment of acute MR as a safe and effective therapeutic alternative to open surgery for high risk patients.
Perel and colleagues reported a case series of patients, who underwent urgent transcutaneous edge to edge repair for severe mitral regurgitation in patients with refractory cardiogenic shock. Despite the relatively small number of reported patients, this cohort is still remarkable, because it reflects the experience of a single centre. The manuscript is compact and straightforward, however there are some missing data, which could provide useful informations for the readers. The authors  referred the heterogeneous etiology of MR and that all patients had previous ischemic heart disease,  presented with an acute ischemic event. As they mentioned “MR severity was diagnosed and evaluated with transthoracic echocardiography (TTE) and transesophageal (TEE)”, however I haven’t found any date on the specific etiology of the MR in the reported cases. Presumably the underlying pathological background (ruptured chorda(e) tendinea(e), papillary muscle dysfunction, partial or complete rupture of the head or the body of papillary muscle, etc.) influences the therapeutic strategy, suitability for  TEER and the prognosis. Similarly, patient selection and exclusion criteria are not exactly defined: “Patients were selected to undergo the procedure due to refractory shock not responsive to standard therapy with concomitant severe MR”. In cases of ischaemic MR emergency coronary revascularisation (PCI) is an integral part of the therapy, which can dramatically reduce the severity of regurgitation and improve the outcome as it  was proved by previous publications. The data referring to the incidence and haemodynamic effect of PCI applied before TEER are missing. This is also the case with the use mechanical circulatory support using intraaortic ballon pump, VA ECMO or other assist devices, which are commonly applied in cases of acute MR complicated with shock and may be life-saving as bridges to recovery or to later therapy. It is not clear whether all inoperable patient were treated with TEER or some of them were declined for any intervention by well defined criteria (for example presence of malignancy with short life expectancy,  futility because of age or condition of a patient, extreme frailty and so, that make transcatheter edge-to-edge repair of prohibitive risk. The authors used New York Heart Association functional class and SCAI stage classification as a marker of severity of patient’s condition and as a predictor of   expected prognosis. In my opinion the use of either of the surgical risk scores (EuroScore, Society of Thoracic Surgeons risk scores) could be a better predictor of mortality and prognosis, moreover it would allow to compare the results of TEER and surgical treatment. The proper timing of intervention in these cases are crucial, which can represent the severity of patient’ condition and influence the prognosis. The time interval from the onset of the ischaemic event or the development of severe MR until the TEER is not documented in the manuscript. Finally, beside the describe postoperative and follow-up data (in-hospital-, 30-day and 6-month mortality, improvement is NYHA class)   the length of ICU and hospital stay as well as the incidence of readmission due to heart failure among patients who underwent transcatheter mitral valve repair would be useful information regarding the early and late outcome  and efficiency of this intervention.                                                                                              Apart from these minor shortcomings of the manuscript, this reflects a huge and very straightforward clinical work, which provides a further step in the evaluation of the role of transcatheter edge-to-edge repair in the treatment of acute MR with CS. Hereby I would like congratulate the authors on their excellent results they achieved in the treatment of this dreadful and devastating clinical scenario! 

Author Response

Thank you for comments. We are truely encouraged by them. 

Regarding MR etiology - All but one patient had MR due to leaflet tethering related to their infarct territory designated as secondary MR. One had a flail leaflet and was designated as primary MR. (Table 1).

Regarding the pre and post echocardiographic evaluation of MR and LV Function - we added Table 2.

Regarding PCI as the primary treatment for these patient - all were evaluated and treated post PCI, expecting improvement that failed to come.

Type of inotropic and mechanical circulatory support was added to table 1.

Regarding the time elapsed in the CCU in full mechanical and medical support before TEER and after TEER - the median were 2 weeks and 1 week respectively.  a sentence describing it was added to results and a paragraph referring to it was added to the discussion section.

Regarding post hospitalization course we have inconsistent information as we lack access to medical records outside our hospital. None were  readmitted to ours, but we cannot be sure whether any of them was admitted to others. the only consistent information was mortality as it comes through the ministry of interior that updates our medical records.

We added Euroscore II to table 1.  Minimum score was 16, highest 37

Reviewer 3 Report

Authors present a cohort of 13 patients with acute MR after ACS treated with MitraClip. Although the manuscript is interesting, I think that authors should improve significantly the results of the manuscript. Was the PCI of the patients performed according to recommendations (emergent primary PCI in STEMI and high risk NSTEMI)? Other important question for me is: why TEER and not mechanical assistant device as bridge to recovery?. If the mechanism was not papillary muscle rupture the improvement of the left ventricle could improve the mitral regurgitation and avoid this procedure. 

In summer, the results must be improved including all the parameters that are important to know the basal status of the patients, the etiology of the mitral regurgitation, the surgical risk,..... to explain the decision of TEER as treatment.

Author Response

  1. Was the PCI of the patients performed according to recommendations (emergent primary PCI in STEMI and high risk NSTEMI)?

Response: We thank the reviewer for this comment.

11 patients have undergone PCI according to current recommendation. 1 patient with NSTEMI presentation has undergone late revascularization (few days from presentation) and 1 patient due to late arrival and severe CKD did not receive revascularization.

8 out of the 13 patients had STEMI and 5 had NSTEMI. 3 patients out of the STEMI group suffered from stent thrombosis, 2 had late arrival and 1 had occlusion of the LIMA graft   

Overall, in most patients, the ischemic MR developed after a large myocardial infarction despite successful revascularization or in patients with incomplete or partially successful revascularization.

We have corrected this segment in the results segment:

All patients presented with an acute ischemic event (STEMI or NSTEMI). 11 patients underwent PCI according to current recommendation, while the remaining 2 patients with a NSTEMI diagnosis have either undergone later revascularization or no revascularization. (3,4) 8 out of the 13 patients included in the study were diagnosed with STEMI and 5 were diagnosed with  NSTEMI. 3 patients from the STEMI group suffered from acute stent thrombosis, 2 had late arrival presentation and 1 suffered from an acute  occlusion of the LIMA graft.   Overall, in most patients, the ischemic MR developed after a large myocardial infarction despite successful revascularization or in patients with incomplete or partially successful revascularization.

  1. Why TEER and not mechanical assistant device as bridge to recovery?. If the mechanism was not papillary muscle rupture the improvement of the left ventricle could improve the mitral regurgitation and avoid this procedure

Response:  Thank you for this comment.

Currently, to the best of our knowledge, there is no robust data on mechanical support versus TEER in patients with cardiogenic shock and severe MR. We agree that patients could benefit from MCS and indeed some of patients received MCS before proceeding with the TEER. Additionally, most of our patients underwent the TEER intervention a few days from the initial ischemic insult and were in severe CS not responding to standard treatment. The decision to procced to TEER was after a shared decision and based on a clinical judgment that those patients needed urgent treatment directly addressing their valvular dysfunction without further waiting for left ventricle improvement. CR Thompson et. al. compared patients with MR versus patients with LVD from the SHOCK Trial Registry. They showed that patients with MR could suffer from the same severity of shock even with better left ventricular function. They highlighted the importance for earlier recognition and potential intervention of the MR in these patients. Their findings support earlier intervention to prevent the vicious cycle of CS and MR.(5)

  1. The results must be improved including all the parameters that are important to know the basal status of the patients, the etiology of the mitral regurgitation, the surgical risk,..... to explain the decision of TEER as treatment.

Response: Thank you again for this comment.

The management of TEER in patients with CS and MR is not guideline-supported and the decision to perform the TEER procedure was made on an individual basis for each patient after a HEART team discussion. As mentioned above, our clinical judgment was to try and address the main valvular problem, and since all patients had a prohibitive surgical risk, the percutaneous option was chosen. All the patients had an acute ischemic event, with 92% of the patients having an ischemic MR not associated with papillary muscle rupture. Haberman et. al. compared conservative, surgical, and percutaneous treatment in patients with mitral regurgitation shortly after acute myocardial infarction. They concluded that early intervention may mitigate the poor prognosis associated with conservative therapy in patients with post-MI MR. Percutaneous mitral valve repair can serve as an alternative for surgery in reducing MR for high-risk patients.(6) In concordance to this trial result, we felt that addressing the clinical scenario with TEER would benefit these high-risk patients and improve their prognosis.

References:

  1. Writing Committee Members, Otto CM, Nishimura RA, Bonow RO, Carabello BA, Erwin JP, et al. 2020 ACC/AHA Guideline for the Management of Patients with Valvular Heart Disease: Executive Summary: A Report of the American College of Cardiology/American Heart Association Joint Committee on Clinical Practice Guidelines. J Am Coll Cardiol [Internet]. 2021 Feb 2 [cited 2021 Dec 2];77(4):450–500. Available from: http://www.ncbi.nlm.nih.gov/pubmed/33342587
  2. Zoghbi WA, Adams D, Bonow RO, Enriquez-Sarano M, Foster E, Grayburn PA, et al. ASE GUIDELINES AND STANDARDS Recommendations for Noninvasive Evaluation of Native Valvular Regurgitation A Report from the American Society of Echocardiography Developed in Collaboration with the Society for Cardiovascular Magnetic Resonance. Journal of the American Society of Echocardiography [Internet]. 2017 [cited 2021 Dec 2];30:303–71. Available from: http://dx.doi.org/10.1016/j.echo.2017.01.007
  3. Collet JP, Thiele H, Barbato E, Barthélémy O, Bauersachs J, Bhatt DL, et al. 2020 ESC Guidelines for the management of acute coronary syndromes in patients presenting without persistent ST-segment elevation. Rev Esp Cardiol (Engl Ed) [Internet]. 2021 Jun 1 [cited 2022 Sep 5];74(6):544. Available from: https://pubmed.ncbi.nlm.nih.gov/34020768/
  4. Ibanez B, James S, Agewall S, Antunes MJ, Bucciarelli-Ducci C, Bueno H, et al. 2017 ESC Guidelines for the management of acute myocardial infarction in patients presenting with ST-segment elevation: The Task Force for the management of acute myocardial infarction in patients presenting with ST-segment elevation of the European Society of Cardiology (ESC). Eur Heart J [Internet]. 2018 Jan 7 [cited 2022 Sep 5];39(2):119–77. Available from: https://pubmed.ncbi.nlm.nih.gov/28886621/
  5. Thompson CR, Buller CE, Sleeper LA, Antonelli TA, Webb JG, Jaber WA, et al. Cardiogenic shock due to acute severe mitral regurgitation complicating acute myocardial infarction: a report from the SHOCK Trial Registry. SHould we use emergently revascularize Occluded Coronaries in cardiogenic shocK? J Am Coll Cardiol [Internet]. 2000 [cited 2022 Sep 5];36(3 Suppl A):1104–9. Available from: https://pubmed.ncbi.nlm.nih.gov/10985712/
  6. Haberman D, Estévez-Loureiro R, Benito-Gonzalez T, Denti P, Arzamendi D, Adamo M, et al. Conservative, surgical, and percutaneous treatment for mitral regurgitation shortly after acute myocardial infarction. Eur Heart J [Internet]. 2022 Feb 14 [cited 2022 Sep 5];43(7):641–50. Available from: https://pubmed.ncbi.nlm.nih.gov/34463727/